# Pyroptosis and Its Role in the Modulation of Cancer Progression and Antitumor Immunity

**DOI:** 10.3390/ijms231810494

**Published:** 2022-09-10

**Authors:** Sihan Qi, Qilin Wang, Junyou Zhang, Qian Liu, Chunyan Li

**Affiliations:** 1School of Engineering Medicine, Beihang University, Beijing 100191, China; 2School of Biological Science and Medical Engineering, Beihang University, Beijing 100191, China; 3Key Laboratory of Big Data-Based Precision Medicine (Ministry of Industry and Information Technology), Beihang University, Beijing 100191, China; 4Beijing Advanced Innovation Center for Big Data-Based Precision Medicine, Beihang University, Beijing 100191, China

**Keywords:** pyroptosis, gasdermin, caspase, programmed cell death, tumorigenesis, immunotherapy

## Abstract

Pyroptosis is a type of programmed cell death (PCD) accompanied by an inflammatory reaction and the rupture of a membrane. Pyroptosis is divided into a canonical pathway triggered by caspase-1, and a non-canonical pathway independent of caspase-1. More and more pyroptosis-related participants, pathways, and regulatory mechanisms have been exploited in recent years. Pyroptosis plays crucial roles in the initiation, progression, and metastasis of cancer and it affects the immunotherapeutic outcome by influencing immune cell infiltration as well. Extensive studies are required to elucidate the molecular mechanisms between pyroptosis and cancer. In this review, we introduce the discovery history of pyroptosis, delineate the signaling pathways of pyroptosis, and then make comparisons between pyroptosis and other types of PCD. Finally, we provide an overview of pyroptosis in different cancer types. With the progression in the field of pyroptosis, new therapeutic targets and strategies can be explored to combat cancer.

## 1. Introduction

Cell death is crucial to maintain in vivo stability during organismal growth and development [1]. Cell death can be divided into two types: accidental cell death and programmed cell death (PCD). Accidental cell death not accurately regulated is caused by harmful physical, chemical, or mechanical stimuli. On the contrary, PCD is a protective suicide to promote morphogenesis and eliminate harmful or abnormal cells [2]. Pyroptosis, also known as cell inflammatory necrosis, is a pro-inflammatory PCD to antagonize infection and endogenous danger signals [3]. Pyroptosis can be divided into canonical pyroptosis, dependent on caspase-1, and non-canonical pyroptosis independent of caspase-1 [4]. Many types of cancer such as non-small cell lung cancer (NSCLC), colorectal cancer (CRC), gastric cancer (GC), and hepatocellular carcinoma (HCC) are associated with inflammation [5]. In addition to cancer, the prevalent role of pyroptosis in other diseases, such as neurodegenerative diseases, has been explored. It has been reported that Huntington’s and Alzheimer’s disease are accompanied by pyroptosis [6,7]. More IL-18 and IL-1β were observed in brain tissues with Huntington’s and Alzheimer’s disease [8,9]. Hence, pyroptosis may contribute to neurodegenerative diseases. A further study on pyroptosis is beneficial to the treatment of neurodegenerative diseases as well. At present, the relationship between pyroptosis and antitumor immunity is unclear, but a number of studies have shown that pyroptosis mediates tumor regression by promoting immune cell activation [10]. Cancer cell pyroptosis (CCP) stimulates inflammatory responses in the tumor microenvironment, which effectively motivates antitumor immunity; immune cell pyroptosis (ICP) is responsible for the host defense against a pathogen infection [11]. Together, both CCP and ICP affect tumor development. As a double-edged sword during carcinogenesis, pyroptosis is still open to discussion. A comprehensive understanding of pyroptosis in the tumor microenvironment will help to develop new and effective therapeutic strategies for cancer.

## 2. The Developmental History of Pyroptosis

Pyroptosis was first observed in 1986 by Friedlander, who noticed an abnormal death phenotype in murine macrophages treated with an anthrax lethal toxin [12]. Caspase-1, which mediates canonical pyroptosis, was discovered to cleave the IL-1β precursor in 1989 [13,14]. Therefore, caspase-1 is also known as an interleukin-1β-converting enzyme. In 1992, the phenomenon of pyroptosis was described as chromatin condensation, cell membrane rupture, endoplasmic reticulum enlargement, and the release of IL-1β as an inflammatory response [15]. Gasdermin, the executor of pyroptosis, was first discovered and named in 2000 [16]. Pyroptosis was regarded as a special form of apoptosis in monocytes until Brennan and Cookson found that macrophages infected with *Salmonella typhimurium* died through an inflammatory death mode different from the traditional form of apoptosis [17]. This pro-inflammatory type of PCD was named pyroptosis in 2001, a word derived from the Greek root, “pyro” [18]. Pyroptosis occurs quickly, destroys the integrity of the cell membrane, and is accompanied by a severe inflammatory process [19]. However, this significant discovery did not attract attention at that time and the specific molecular mechanism of pyroptosis remains unclear. One year later, the term “inflammasome” was proposed to replace the caspase-activating complex, which activates the inflammatory caspases [20].

With further exploration, the definition of pyroptosis and related studies also continuously evolve and are updated. In 2009, the Nomenclature Committee on Cell Death (NCCD) redefined pyroptosis as inflammatory cell death caused by the activation of caspase-1 [21]. It has been gradually realized that in addition to caspase-1, caspase-4/5/11, granzyme A (GZMA), and granzyme B (GZMB) could also cause pyroptosis. Gasdermin D (GSDMD), a gasdermin family member, was identified as the substrate of inflammatory caspase-1 through an enzymatic N-terminal enrichment method with mass spectrometry-based proteomics in 2010 [22]. Furthermore, other members of the caspase family and gasdermin family were also found to participate in pyroptosis. Caspase-4/5/11 was found to induce pyroptosis independently of caspase-1 by directly identifying the lipopolysaccharide (LPS) to activate GSDMD in 2014 [23]. One year later, a breakthrough work on pyroptosis demonstrated that the gasdermin protein family was the direct executor of pyroptosis [24]. This is another key substrate of inflammatory caspases in addition to IL-1β/IL-18. Subsequently, in 2016, several studies demonstrated that after cleavage by inflammatory caspases and the disruption of the self-inhibitory state of GSDMD, GSDMD-N peptides were released to bind to the membrane lipids, resulting in membrane disruption and pyroptosis [25,26,27]. Since then, the importance of the gasdermin family during pyroptosis has been gradually realized. In 2018, the NCCD revised pyroptosis as a type of cell death dependent on the gasdermin family to punch holes in the membrane mostly activated by an inflammatory caspase [28]. In recent years, granzymes such as GZMA and GZMB have been discovered to cause pyroptosis by cleaving gasdermin B (GSDMB) and gasdermin E (GSDME), respectively [29,30]. The cognition and definition of pyroptosis are still evolving. Several crucial time points for the discovery of pyroptosis are depicted in Figure 1.

## 3. The Process of Pyroptosis

A pathogen invasion into phagocytic cells can be detected by inflammasomes, which induce pyroptosis by the release of inflammatory markers such as IL-18 and IL-1β. Phagocytes are then recruited to kill the cells infected by the pathogen [31]. Pyroptosis is generally divided into two main categories: the canonical pathway and the non-canonical pathway. Canonical pyroptosis, the earliest discovered form of pyroptosis, is mediated by caspase-1. Non-canonical pyroptosis describes the rest of the types other than canonical pyroptosis. Generally speaking, the process of pyroptosis can be divided into four main phases (Figure 2): (1) the capture of stimulatory signals; (2) the transmission of stimulatory signals; (3) the activation of pyroptosis executors; and (4) the execution of pyroptosis.

### 3.1. The Capture of Stimulatory Signals

Pyroptosis, as an important innate immune response in the body, plays a significant role in antagonizing infection and endogenous danger-signaling processes [32]. Pattern recognition receptors (PRRs) mediate the occurrence of innate immunity by recognizing pathogen-associated molecular patterns (PAMPs) and damage-associated molecular patterns (DAMPs) [33]. PAMPs are conserved structures in pathogenic microorganisms, including nucleic acid, surface glycoprotein, lipoprotein, and membrane components [34]. DAMPs, which are endogenous molecules derived from damaged cells or tissues, can trigger the immune responses of the body [35,36].

The acquisition of a stimulatory signal depends on the PRRs in the canonical pathway whereas the recognition of the non-canonical pyroptosis signal is diverse. The capture of the pyroptosis signal is different among different Gram-negative bacteria. The *Yersinia* virulence factor YopJ enters the host cells via the type III secretion system whereas LPS is transported into the host cells by endocytosis via secreted outer membrane vesicles (OMVs) for a few other Gram-negative bacteria such as enterohaemorrhagic *E. coli* [37,38,39,40,41]. The granzyme, a serine protease produced by natural killer cells and cytotoxic T cells, is delivered by perforin to the target cells [29,42,43]. Moreover, chemotherapy drugs such as cisplatin can be transported to cells through the passive diffusion of the plasma membrane and copper transporter protein CTR1 to induce pyroptosis [44,45,46].

### 3.2. The Transmission of Stimulatory Signals 

Once stimulation is received by the PRRs, the signal will be further transmitted to the inflammasomes to mediate the canonical pyroptosis pathway (Figure 2). Inflammasome, a multi-protein complex to activate caspase, is composed of receptor proteins, apoptosis-associated speck-like protein-containing CARD (ASC), and effector protein pro-caspase-1 [20,47,48]. Receptor proteins include nod-like receptors, absent in melanoma 2 (AIM2)-like receptors, and melanoma absent factor 2 [49]. Nucleotide-binding oligomerization domain, leucine-rich repeat, and pyrin domain-containing 3 (NLRP3) are the most characteristic inflammasomes in nod-like receptors [50]. After the capture of danger signals through the PRRs, NLRP3 oligomerizes and interacts with the PYD domain of ASC and then the adaptor ASC recruits cystine protein pro-caspase-1 to produce active caspase-1, which triggers a series of subsequent pyrolytic reactions [51]. NIMA-related kinase 7 (Nek7) is the central regulator of the NLRP3 inflammasome; the loss of *Nek7* protected macrophages from nigericin-induced pyroptosis in mice [52].

For the non-canonical pyroptosis pathway, the transmission of stimulatory signals is diverse. YopJ inhibits TGFβ-activated kinase-1 (TAK1) to activate caspase-8 as soon as YopJ enters the cell [53]. LPS activates caspase-11 to transmit the pyroptosis signal [54]. In addition, GZMA and GZMB from cytotoxic lymphocytes (such as cytotoxic T lymphocytes and natural killer cells) enter the target cells through perforin to induce pyroptosis [29,30]. Chemotherapeutic drugs stimulate pyroptosis by activating caspase-3 [44].

### 3.3. The Activation of Pyroptosis Executors

When pyroptosis was officially recognized as a new type of PCD, only caspase-1 was identified to activate IL-1β after an infection and cause cell death. Pyroptosis was initially defined as “inflammatory cell death caused by activation of caspase-1” in 2009 [21]. Caspases are divided into pro-apoptotic and pro-inflammatory caspases. Pro-apoptotic caspases, which are mainly involved in apoptosis, are classified into initiator caspases (caspase-8, -9, and -10) and executor caspases (caspase-3, -6, and -7) [55]. Initiator caspases activate the executor caspase to trigger the occurrence of apoptosis. Pro-inflammatory caspases are composed of caspase-1, -11, and -12 in mice and caspase-1, -4, and -5 in humans [56]. For example, once caspase-11 was activated by LPS, caspase-11 induced pyroptosis to protect cells from a lethal infection by the Gram-negative bacteria *B. thalandensis* and *B. pseudomallei* in mouse macrophages [57,58]. These series of studies on caspase-11 proposed the existence of non-canonical pyroptosis and also switched the focus to other inflammatory caspases. The apoptosis mediated by caspase-3 can be converted to pyroptosis by tumor necrosis factor or chemotherapy drugs [44]. The activated caspase-3 cleaves GSDME at the Asp270 site to release a GSDME-N fragment; GSDME-N destroys the integrity of the cell membrane by punching holes in the cell membrane and then the inflammatory factors are discharged outside the cell [59]. YopJ, an effector molecule produced by *Yersinia pestis*, activates caspase-8 by inhibiting TAK1–IκB kinase signaling; the activated caspase-8 then triggers pyroptosis by cleaving GSDMD [41,53]. The findings above break the concept that caspase-3 and caspase-8 activation are unique to apoptosis and further expand the understanding of pyroptosis.

In addition to caspases, serine protease granzymes can activate pyroptosis executors, as well. For example, GZMA cleaved GSDMB molecules at the Lys^229^/Lys^244^ site to activate GSDMB and induce pyroptosis in target cells [29]. Another serine protease, GZMB, directly cleaved GSDME to induce GSDME-dependent pyroptosis in HeLa cells (Figure 2) [30]. These findings break the view that pyroptosis can only be activated by caspases.

### 3.4. The Execution of Pyroptosis

The downstream gasdermin protein determines the occurrence of pyroptosis. Gasdermin originated from Gsdmal in mouse gastrointestinal and skin epithelial cells [16]. There are six homologs of the human gasdermin (GSDM) family: GSDMA, GSDMB, GSDMC, GSDMD, GSDME, and DFNB59 [60]. When GSDMs are cleaved by proteases such as caspase, the autoinhibited conformation formed by GSDMN and GSDMC is broken; the GSDMN termini then perforate the cell membrane by targeting phosphoinositides and cardiolipin, resulting in cell swelling and lysis [61,62]. In brief, GSDMs have no effect on the production and maturation of inflammatory factors, but promote the release of inflammatory factors.

The mechanism of GSDMs in pyroptosis is being constantly investigated. Although GSDMA is capable of making holes in the cell membrane, there is currently no evidence to demonstrate that GSDMA is associated with pyroptosis [63]. GSDMB is associated with human immune diseases and involved in non-canonical pyroptosis via caspase-4 or GZMA [29,64]. GSDMC was initially detected in the epithelial cells of the upper digestive tract [16]. α-ketoglutarate, a metabolite of the tricarboxylic acid cycle, triggers pyroptosis by activating GSDMC via caspase-8 [65]. Hypoxia induces PD-L1 to translocate to the nucleus and bind to phosphorylated STAT3, which switches tumor necrosis factor α-mediated apoptosis to pyroptosis via GSDMC [66]. GSDMD is the common substrate of caspase-1 and caspase-4/5/11. In 2015, two research teams found that GSDMD was a direct target of inflammatory caspase [24,67]. The activated caspase cleaves the junction region of the N-terminal and C-terminal domains of the GSDMD protein to release the N-terminal domain, which binds to the membrane phospholipids and then destroys the cell membrane to induce pyroptosis [61,68]. The endosomal sorting complex required for transport (ESCRT) is regulated by a Ca^2+^ influx and repairs the plasma membrane by contracting the damaged plasma membrane through molecules such as CHMP4B [69,70,71]. ESCRT reduces pyroptosis by repairing the membrane pores formed by GSDMD [72]. Moreover, Mg^2+^ blocks the Ca^2+^ influx by inhibiting the Ca^2+^ channel P2RX7 and then restrains LPS–caspase-11–GSDMD-mediated pyroptosis [73]. The above-mentioned negative regulation of pyroptosis eliminates excessive inflammation caused by pathogens and maintains immune homeostasis after an infection. GSDME, also named “deafness autosomal dominant 5”, was initially identified as a dominant gene associated with progressive hearing loss [74]. Caspase-3 cleaves GSDME to release the N-terminal of GSDME, which perforates the membrane and leads to pyroptosis [75]. Together, the gasdermin family is the ultimate executor of pyroptosis. A more comprehensive understanding of GSDMs is required to explore the underlying regulatory mechanisms of pyroptosis.

## 4. Three Categories of PCD: Apoptosis, Pyroptosis, and Ferroptosis

PCD removes senescent, redundant, and potentially tumorigenic cells, which is indispensable to maintain the homeostasis of growth and development. PCD is crucial to the regulation of the host defense against pathogens [76]. Apoptosis is the earliest and most classical mode of PCD; other types of PCD have been subsequently defined such as pyroptosis and ferroptosis. The abnormality of PCD is associated with the occurrence and development of cancer. For example, the downregulation of the tumor suppressor p53 leads to a decrease in apoptosis, resulting in enhanced tumor growth and progression [77]. Resistance to apoptosis and an immunosuppressive tumor microenvironment are two major causes of a poor response.

Apoptosis removes defective and damaged cells to ensure the health of the organism under physiological conditions [78]. Apoptosis ubiquitously occurs in the process of tissue remodeling, biological growth, and development [79]. In Greek, apoptosis means that petals fall from flowers and leaves fall from trees [80]. Once apoptosis occurs, there are distinct characteristics in the cell morphology. First, chromatin condensation and nuclear fragmentation occur. The surface of cell membrane then sprouts and forms apoptotic bodies, which contain an intact cytoplasm with a complete membrane, organelles, and nuclear fragments. Eventually, the apoptotic cells are devoured by phagocytes [81]. As macrophages promptly clear apoptotic cells, the occurrence of apoptosis is not accompanied by inflammation to minimize the impact on surrounding cells, which is different from pyroptosis and ferroptosis. Caspase is the key component in the process of apoptosis. For apoptosis, caspases are functionally subdivided into initiator caspases (caspase-8/9/10) and effector caspases (caspase-3/6/7) [55]. In pyroptosis, caspase-1 mediates the canonical pathway; caspase-4/5/11 is involved in the non-canonical pathway [24].

Unlike apoptosis, pyroptosis occurs with strong inflammatory responses and is crucial in innate immunity against pathogens [82,83]. Both pyroptosis and apoptosis concomitantly occur with chromatin condensation. The cell membrane is intact for apoptosis; on the contrary, for pyroptosis, there are membrane disruptions, cell volume expansion, subsequent content efflux, and then inflammatory reactions [3,84]. Caspases associated with pyroptosis are called inflammatory caspases and include caspase-1/4/5/11 [31]. Different from apoptosis, caspases are not necessary for pyroptosis. Gasdermins are the core components of pyroptosis and cleaved gasdermins are major executors of pyroptosis [62]. Both apoptosis and pyroptosis are involved in the establishment of the cancer microenvironment to modulate cancer progression and therapeutic responses [85].

Ferroptosis, caused by iron-dependent lipid peroxidation, was first defined by Dixon et al. in 2012 [86]. Distinct from apoptosis and pyroptosis, ferroptosis occurs without chromatin condensation and does not require caspase [87]. Mitochondrial atrophy is a representative morphological feature of ferroptosis [86]. Consistent with pyroptosis, the nucleus is intact and inflammatory responses occur, along with ferroptosis [88]. Glutathione peroxidase 4 (GPX4), a key component in the regulation of ferroptosis, maintains metabolic homeostasis by dissipating lipid peroxides [89,90]. Ferroptosis occurs if the expression of GPX4 is inhibited [91]. In cancer cells, the metabolic rate, level of reactive oxygen species, and iron content are higher than those of normal cells [92]. Based on the characteristics above, ferroptosis in cancer cells can suppress tumor growth [93]. Thus, inducing ferroptosis in cancer cells may be a potential therapeutic approach.

To summarize, a comparison of the three types of PCD are depicted in Table 1. Investigations into the molecular mechanisms of these three types of PCD during carcinogenesis may be an approach for cancer therapy in the future.

## 5. Pyroptosis in Cancer Progression and Chemotherapeutic Responses

Pyroptosis, an inflammatory cell death, plays an essential role in immunity as well as cancer progression [94]. Pyroptosis in different microenvironments may have diverse effects on carcinogenesis and induce completely opposite outcomes in cancer therapy. Here, we review the function of pyroptosis in the main cancer types; we summarize the function of pyroptosis in the main cancer types in Table 2.

### 5.1. Pyroptosis in NSCLC and Its Chemotherapeutic Responses

Lung cancer is the most common type of cancer worldwide and approximately 85% of lung cancer is NSCLC [95]. 4-hydroxybenzoic acid induces pyroptosis in A549 cells through the caspase-1/IL-1β pathway, resulting in NSCLC growth inhibition [96]. Simvastatin, as a statin with anticancer properties, may be applied in the treatment of NSCLC [97]. The activation of the NLRP3 inflammasome and caspase-1 by simvastatin stimulates pyroptosis via the canonical pathway, resulting in the migration inhibition of NSCLC (Figure 3A) [98]. Polyphyllin VI, a component isolated from *Trillium tschonoskii* Maxim, inhibits NSCLC development by inducing pyroptosis via the activation of the NLRP3 inflammasome–caspase-1–GSDMD pathway (Figure 3A) [99,100]. The expression of *GSDME* is significantly reduced in lung cancer tissue compared with normal tissue; in addition, patients with a low expression of *GSDME* presented a poor prognosis under cisplatin treatment [101]. Therefore, GSDME may serve as a prognostic marker for a personalized therapy.

### 5.2. Pyroptosis in HCC and Its Chemotherapeutic Responses

HCC, as one of the most prevalent malignancies, often results from chronic hepatitis and cirrhosis [102]. Chemotherapy and immunotherapy for advanced HCC have limited efficacy nowadays [103]. A differential expression analysis of 33 pyroptosis-related genes (PRGs) between normal liver and HCC samples from The Cancer Genome Atlas (TCGA) database demonstrated that only 3 of the 26 differentially expressed genes were significantly downregulated; the remaining 23 differentially expressed genes were significantly upregulated [104]. As most PRGs are upregulated, PRGs may be explored as prognostic biomarkers for HCC. Sorafenib is a kinase inhibitor that achieves a therapeutic effect for HCC by modulating the tumor microenvironment [105,106]. In addition to the direct effect on cancer cells and angiogenesis, other immunomodulatory effects of sorafenib have recently been reported. Sorafenib induces the macrophage to undergo pyroptosis and release pro-inflammatory cytokine; natural killer (NK) cells are then activated to ultimately eliminate the hepatocellular cancer cells (Figure 3B) [107]. Alpinumisoflavone induces NLRP3-mediated pyroptosis to inhibit the proliferation of SMMC 7721 and Huh7 cells, resulting in HCC cell-growth suppression [108]. Berberine, an isoquinoline quaternary alkaloid isolated from medicinal plants, inhibits the proliferation and migration of cancer cells [109]. Berberine induces pyroptosis in HepG2 cells by promoting the expression of caspase-1, inhibiting the migration and proliferation ability of HepG2 cells; and in vivo experiments showed that the tumor volume was significantly shrinked after Berberine treatment compared with that of the control group (Figure 3B) [110]. In addition, a high expression of GSDME is significantly correlated with a short overall survival whereas other GSDMs are not [111].

### 5.3. Pyroptosis in CRC and Its Chemotherapeutic Responses

CRC, a digestive malignancy with a high morbidity, is mainly caused by chronic inflammation [112]. LPS from the outer membrane of Gram-negative bacteria improves the sensitivity of CRC to oxaliplatin and increases antitumor activity by inducing GSDMD-mediated pyroptosis in HT-29 cells [113]. Camptothecin analogue FL118 inhibits CRC growth and metastasis by inducing NLRP3/caspase-1-mediated pyroptosis in SW48 and HT129 cells [114]. The antitumor drug 5-aza-2-deoxycytidine, a DNA methylation inhibitor, treats CRC by upregulating the expression of *NLRP1*. The expression levels of *NLRP1* were increased after both in vitro and in vivo treatments by DAC, resulting in CRC inhibition [115]. Therefore, the NLRP1 inflammasome is a negative regulator of intestinal tumorigenesis. In *NLRP3*-deficient mice, the incidence of CRC tended to increase, indicating that NLRP3 is also a negative regulator of intestinal tumorigenesis [116]. Another study also showed that the *AIM2* expression was absent in nearly two-thirds of CRC patients and the loss of the *AIM2* expression may be an important biomarker to evaluate and identify CRC patients with a poor prognosis [117]. In addition, the knockout of *TGFBR2* in CRC resulted in the upregulation of the *GSDMC* expression and promoted the proliferation of tumor cells [118]. *GSDMC* plays an oncogene role in the occurrence of CRC; *GSDME* is a tumor suppressor gene, which may serve as a biomarker for a CRC diagnosis [119]. In HT-29 and HCT116 cells, lobaplatin induces pyroptosis via the activation of caspase-3 and GSDME, which provided evidence that lobaplatin eradicate CRC cells via proptosis (Figure 3C) [120].

### 5.4. Pyroptosis in GC and Its Chemotherapeutic Responses

GC has a high mortality and recurrence rate and is mainly caused by an infection of *H. pylori* [121,122]. There is no specific chemotherapy for GC as yet. A total of 11 pyroptosis-related regulators, including *CASP1*, *CASP3*, *CASP4*, *CASP5*, *CASP8*, *GSDMB*, *GSMDC*, *GSDMD*, *GSDME*, *GZMA*, and *GZMB*, are highly expressed in GC [123]. This indicates that the expression levels of pyroptosis-related regulatory genes are closely associated with GC; this provides a new strategy to predict the survival and prognosis of GC patients from the perspective of pyroptosis. *GSDMB* and *GSDMC* are considered to be tumor suppressor genes in GC [124]. BIX-01294, with chemotherapeutic agent cisplatin, induced caspase-3/GSDMD-mediated pyroptosis in SGC-7901 cells and restrained GC growth [125]. A 5-fluorouracil treatment induced the expression of *GSDME*, which switched caspase-3-dependent apoptosis to pyroptosis (Figure 3D) [126]. Another study reported that a loss of the *GSDME* expression promoted tumor cell growth *in vivo* and *in vitro* [127]. Therefore, *GSDME*, as a tumor suppressor gene in GC, may be explored as a therapeutic target for GC by inducing pyroptosis.

To date, most studies have focused on chemical drug-induced pyroptosis, but drug resistance and severe side effects hinder their application. New approaches are being explored to induce pyroptosis to stimulate cancer therapies such as photon-mediated biomedical engineering techniques [128].

**Table 2 ijms-23-10494-t002:** The effect of pyroptosis on cancer in the main cancer types.

Cancer Type	Pyroptosis-Related Genes	Effect on Cancer	References
NSCLC	Caspase-1	Inhibit	[96,98,100]
	NLRP3	Inhibit	[98,100]
	GSDMD	Inhibit	[100]
	GSDME	Inhibit	[101]
HCC	Caspase-1	Inhibit	[110]
	NLRP3	Inhibit	[108]
	GSDME	Promote	[111]
CRC	Caspase-1	Inhibit	[114]
	Caspase-3	Inhibit	[120]
	NLRP1	Inhibit	[115]
	NLRP3	Inhibit	[114,116]
	AIM2	Inhibit	[117]
	GSDMC	Promote	[118]
	GSDME	Inhibit	[119,120]
GC	Caspase-3	Inhibit	[125]
	GSDMB, GSBMC	Inhibit	[124]
	GSDMD	Inhibit	[125]
	GSDME	Inhibit	[127]

## 6. Pyroptosis in Immunotherapy

Chemotherapy and radiotherapy are conventional cancer treatment methods. However, these treatments quickly kill both cancer cells and normal cells, including immune cells [129]. Immunotherapy stimulates the immune system to eliminate cancer cells [130]. An immune checkpoint inhibitor (ICI) therapy, particularly anti-PD1 and anti-PD-L1, primarily acts by activating pre-existing tumor immune responses [131]. PD-L1, caused by hypoxia, enters the nucleus, binds to phosphorylated STAT3 (which forms a complex with the promoter region of GSDMC), and then triggers pyroptosis [66]. In summary, GSDMC-mediated CCP promotes tumor necrosis and restrains tumor development. Perforin and granzyme are two protein toxins released by cytotoxic lymphocytes to kill cancer cells; perforin released by CTLs enabled a tumor regression by inducing pyroptosis through GZMA [29,42]. Cancer cells can escape the “hunt” of the immune system by resorting to ESCRT-mediated membrane repair. The inhibition of the ESCRT pathway by *CHMP4B* knockout increased the killing power of CTLs against cancer cells [132]. In addition, our research group established a novel strategy to predict cancer patient survival and immunotherapy outcomes from the perspective of pyroptosis and screened out five PRGs that may further enhance immunotherapy [133]. On the one hand, pyroptosis alters the tumor microenvironment and inhibits tumor growth by releasing inflammatory factors such as IL-1β and IL-18; on the other hand, pyroptosis reduces the immune responses of the body to tumor cells and accelerates the growth rate of different cancers [134]. Therefore, how to balance the two requires a further understanding of pyroptosis in cancer progression and the anticancer potential.

## 7. Conclusions

Pyroptosis, different from the other forms of PCDs (apoptosis and ferroptosis), is accompanied by a cell rupture and severe inflammatory reaction. Here, we provided a comprehensive introduction to pyroptosis and discussed the relationship between pyroptosis and carcinogenesis. First, there are various stimulation signals to activate pyroptosis, including PAMPs, DAMPs, drug stimulations, and granzymes. Most of the stimuli are transmitted by activating caspases and granzymes, which subsequently activate gasdermins. Gasdermins are then cleaved by active caspases or granzymes to expose the gasdermin N-terminus, which punch the cell membrane. Finally, pyroptosis induces the release of intracellular inflammatory factors IL-1β and IL-18 to trigger inflammation and cell death. Increasingly, studies have shown that pyroptosis is closely associated with cancer.

However, the relationship between pyroptosis and cancer is not well-defined at present. On one hand, the occurrence of pyroptosis can effectively regulate the tumor immune microenvironment, activate a strong T cell-mediated antitumor immune response, inhibit tumor growth, and enhance the sensitivity of cancer cells to chemotherapeutic drugs. On the other hand, as pyroptosis is a pro-inflammatory cell death mode, it also provides a suitable microenvironment for tumor growth. Therefore, studies on the mechanism of pyroptosis can provide new strategies for follow-up cancer treatments. The link between pyroptosis and tumor immunity also provides important ideas for cancer treatments; pyroptosis plays a driving role in tumor immunity and tumor immunotherapy. As increasing studies have shown that pyroptosis is a “divine assistant” for immunotherapy, the development of pyroptosis-related agonists will potentially become a booster to enhance the immunotherapeutic efficacy.

## Figures and Tables

**Figure 1 ijms-23-10494-f001:**
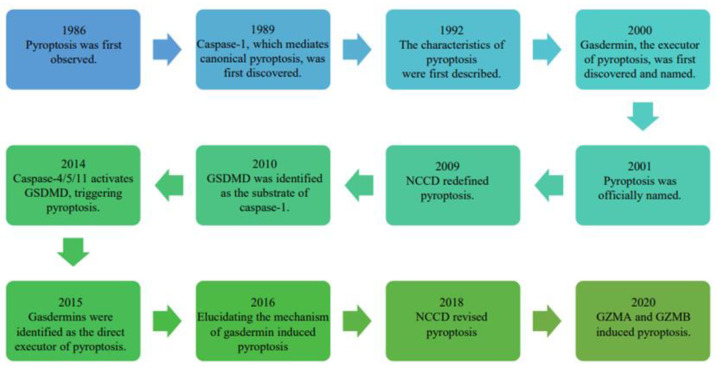
The timeline for the study of pyroptosis. The nodes represent the important events since the first observation in 1986 to the present day.

**Figure 2 ijms-23-10494-f002:**
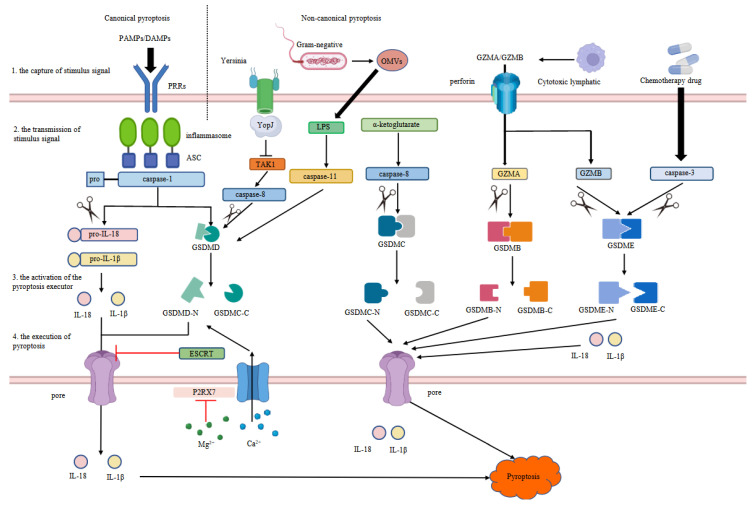
The molecular mechanisms of pyroptosis. Both canonical and non-canonical pathways can be divided into four stages: (1) the capture of stimulus signal; (2) the transmission of stimulus signal; (3) the activation of the pyroptosis executor; and (4) the execution of pyroptosis.

**Figure 3 ijms-23-10494-f003:**
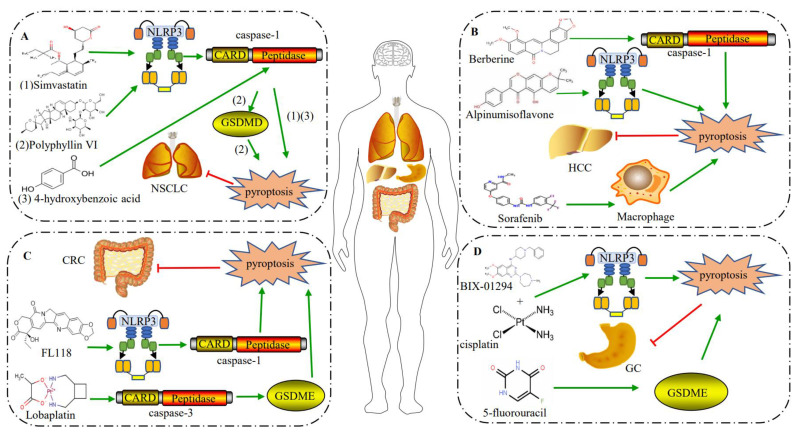
Pyroptosis induced by chemotherapeutic agents for cancer therapy. (**A**) Simvastatin (1), polyphyllin VI (2), and 4-hydroxybenzoic acid (3) inhibit NSCLC by promoting the expression of NLRP3 inflammasome and caspase-1 to induce pyroptosis. (**B**) Berberine, Alpinumisoflavone, and sorafenib induce pyroptosis to restrain HCC. (**C**) FL118 and Lobaplatin induces pyroptosis via NLRP3–caspase-1 and caspase-3–DSDME to suppress CRC, respectively. (**D**) To cure GC, BIX-01294 with cisplatin and 5-fluorouracil induce pyroptosis through NLRP3 and GSDME, respectively.

**Table 1 ijms-23-10494-t001:** Comparison of the three types of PCD.

	Apoptosis	Pyroptosis	Ferroptosis
Cell death mode	PCD	PCD	PCD
Induced factors	Gene regulation under physiological conditions	Pathological stimulus	Lipid peroxidation
Chromatin	Condensation	Condensation	Non-condensation
Nucleus	Fracture	Complete	Complete
Inflammatory response	No	Yes	Yes
Morphological characteristic	Apoptotic bodies	Cell swelling	Mitochondrial atrophy
Initiator caspase	Caspase-8/9/10	Caspase-1/4/5/11	Independent of caspase
Key component	Caspase	Gasdermin	GPX4

## Data Availability

Not applicable.

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
