# Peer review of "Pyroptosis and Its Role in the Modulation of Cancer Progression and Antitumor Immunity"

_ijms, 2022, doi:10.3390/ijms231810494_

Round 1

Reviewer 1 Report

This is a detailed overview of the process of pyroptosis and its associations with cancer. The information is comprehensive, and the figures are well-placed and informative. However, the review could use more organization to improve the readability. A few suggestions that could help :

1. A table summarizing the roles of pyroptosis in the different cancer types, along with list of studies/references would be helpful.

2. A short paragraph on the roles of pyroptosis in other diseases, including Huntington's and neuro degeneration would be timely to add

3. Figure 2 is a bit complicated and hard to read- it would be good for the authors to revise this figure, or possibly break it up into smaller panels

Reviewer 2 Report

The manuscript by Qi and colleagues recapitulates the mechanisms of pyroptosis and the role played by this programmed cell death n antitumor immunity and cancer progression. The manuscript is clear, well written and easy to follow.    

Manuscript’s strengths

1-    the manuscript clearly recapitulates the molecular mechanisms involved in pyroptosis

2-    the manuscript clearly describes the involvement of pyroptosis in cancer

3-    the manuscript clearly describes how pyroptosis induction represents a useful tool for fighting cancer

4-    the references considered are updated and in line with the topic

Manuscript’s weaknesses

No important flaws identified. The quality (resolution and dimension) of the figures can be improved.

Reviewer 3 Report

In the manuscript “Pyroptosis and its role in the modulation of cancer progression and antitumor immunity”  Qi, S., et al have introduced the pyroptosis to the readers, and then mentioned about various pathways regulating pyroptosis.  In addition, the authors have covered the role of pyroptosis in the initiation, progression and metastatic spread of cancer cells.  Although interesting, the article in its current form requires thorough revision before considering it for further action.   

Following are the major comments

1.     There are many articles published on Pyroptosis and its role in cancer and anti-tumor immunity.  The authors should explain how different this article is from already published reviews and mention the need to have one more article in the similar line.  Some of the recently published articles are listed below.  Authors can go through these and highlight the key research gaps in this area.

2.     Authors would have focused more on therapeutic aspects, in particular the pharmacological agents targeting Pyroptosis

3.     A conclusion message with future directions should be provided

Following are some of the articles recently published:

1. REVIEW article

Front. Oncol., 26 July 2021.  Pyroptosis, a New Breakthrough in Cancer Treatment

Wu, D., et al

2. Front Oncol. 2021; 11: 635774. Induction of Pyroptosis: A Promising Strategy for Cancer Treatment.  Lei Wang, 1 Xiaowei Qin, 1 Jianmin Liang, 2 , * and Pengfei Ge 1 , *

3. Cancers (Basel). 2021 Jul; 13(14): 3620. Pyroptosis in Cancer: Friend or Foe?

Xiuxia Lu, Tianhui Guo, and Xing Zhang*

4. Review Article. Genes & Diseases.  The implication of pyroptosis in cancer immunology: Current advances and prospects Wei Liu, Jin wu Peng, Muzhang Xiao, YuanCai, BiPeng, Wenqin, Zhang, JianboLi, Fanhua Kang, Qianhui Hong, Qiuju Liang,, Yuanliang Yan, Zhijie Xu. Available online 12 May 2022 In Press, Corrected Proof

5. Cancer Biology & Medicine:  Methods for monitoring cancer cell pyroptosis

Shuo Wang, Yuantong Liu, Lu Zhang and Zhijun Sun April 2022, 19 (4) 398-414

6. Theranostics 2022; 12(9): 4310-4329 Pyroptosis in inflammatory diseases and cancer Zhiping Rao, Yutong Zhu, Peng Yang, Zhuang Chen, Yuqiong Xia, Chaoqiang Qiao, Weijing Liu, Hongzhang Deng, Jianxiong Li, Pengbo Ning, Zhongliang Wang

7. Photocatalytic Superoxide Radical Generator that Induces Pyroptosis in Cancer Cells

Le Yu, Yunjie Xu, Zhongji Pu, Heemin Kang, Mingle Li, Jonathan L. Sessler, and, Jong Seung Kim J. Am. Chem. Soc. 2022, 144, 25, 11326–11337.  Publication Date: June 16, 2022

Minor comments:

Line #165:  Hela should be written as HeLa.

Round 2

Reviewer 3 Report

Authors have addressed the queries to the satisfaction